# Effectiveness of a Family Intervention to Increase Physical Activity in Disadvantaged Areas—A Healthy Generation, a Controlled Pilot Study

**DOI:** 10.3390/ijerph17113794

**Published:** 2020-05-27

**Authors:** Gisela Nyberg, Susanne Andermo, Anja Nordenfelt, Matthias Lidin, Mai-Lis Hellénius

**Affiliations:** 1Department of Global Public Health, Karolinska Institutet, 171 77 Stockholm, Sweden; susanne.andermo@ki.se; 2The Swedish School of Sport and Health Sciences (GIH), 114 86 Stockholm, Sweden; 3Department of Neurobiology, Care Sciences and Society, Karolinska Institutet, 141 83 Huddinge, Sweden; 4The Foundation A Healthy Generation, 118 63 Stockholm, Sweden; anja@nordenfelt.se; 5Department of Medicine, Karolinska Institutet, 171 76 Stockholm, Sweden; matthias.lidin@ki.se (M.L.); Mai-Lis.Hellenius@ki.se (M.-L.H.); 6Theme Heart and Vessel, Karolinska University Hospital, 171 76 Stockholm, Sweden

**Keywords:** mothers, fathers, school, organized physical activity, healthy meals, health information, parental support groups, accelerometers

## Abstract

There are large social inequalities in health. The purpose of this study was to evaluate the effects of a family intervention on physical activity (PA) and sedentary time (ST) in children and their parents. In this controlled pilot study, all 8–9-year-old children from four schools from a socioeconomically disadvantaged area in Sweden were invited and 67 children and 94 parents were included. The intervention was run by a foundation in co-operation with the municipality. The 9-month program included: (1) activity sessions, (2) healthy meals, (3) health information and (4) parental support groups. PA was primary outcome and ST was secondary outcome, measured by accelerometry. In total, 40 of the children (60%) and 45 of the adults (50%) had at least one day of valid accelerometer data at both baseline and follow-up. Significant intervention effects for the whole group were found in total PA (*p* = 0.048, mean difference (MD) intervention/control 150 counts per minute) and in vigorous PA (*p* = 0.02, MD 8 min/day) during the weekends. There were no differences between groups in the other PA variables or ST. This pilot study shows that it is possible to influence PA in families from a disadvantaged area through a family program.

## 1. Introduction

Being physically active during childhood and adolescence can lead to significant health benefits, both in the short and long term [1,2]. In Sweden, it has been shown that only 23% of girls and 43% of boys aged 11–18 years complied to the physical activity recommendation of at least 60 min moderate to vigorous physical activity (MVPA) per day [3] and the same situation was also evident worldwide [4,5,6]. Additionally, girls were less physically active than boys and the physical activity levels declined, for both girls and boys, with increasing age [7]. Previous studies from Sweden [8] and elsewhere [9] showed that children and adolescents were less physically active during leisure time compared to school time and during weekdays compared to weekends. In addition, physical activity levels have been shown to decrease more during weekends compared to weekdays in children [10]. The mean daily sedentary time among children has been shown to be 8.6 h and girls were more sedentary than boys [11]. In adults it was shown in a large global study, using questionnaires, that more than 25% were insufficiently physically active [12]. In Sweden it was shown with accelerometer measurements that only 7% of adults reached the recommended 150 min of MVPA per week [13].

There are large social inequalities, seen already from an early age, in physical activity and prevalence of obesity to the disadvantage of children from families with low socioeconomic status (SES) [14,15]. Furthermore, inequalities in health are increasing between socioeconomic groups in Sweden [16]. There is evidence to suggest that health-related behaviors [17,18] and obesity track to a certain degree from childhood to adolescence and adulthood [19], which in later life may result in serious health consequences, such as metabolic disturbances, type 2-diabetes, cardiovascular diseases, certain cancers and impaired mobility [20]. Importantly, studies have also shown that children who have a physically active lifestyle keep an active lifestyle into adulthood [21,22]. Therefore, interventions should start at an early age.

Numerous interventions for increasing physical activity in children, especially in the school setting, have been carried out showing small effects [23]. However, interventions tend to be more successful when there is a home-element and parents involved [24,25]. Parents are important in influencing their children’s physical activity and health behaviors and it has been shown in a longitudinal study that higher levels of parental physical activity were associated with increased levels of physical activity among their offspring from childhood to middle age [26]. Previous systematic reviews [27,28] studying the effect of family interventions have shown small but significant effects on increasing physical activity and reducing sedentary behavior and therefore family interventions might be a promising way to increase physical activity and reduce sedentary behavior in children. 

The foundation A Healthy Generation was launched in 2011, with the aim to increase physical activity and promote a healthy lifestyle, in children and their families in disadvantaged areas in Sweden. In close cooperation with municipalities and local organizations, the foundation runs their program for the duration of one school year. The municipalities sign a contract and employ qualified health coordinators that are trained by the foundation. To date, the program has been implemented in 10 municipalities and 17 areas around Sweden and more than 1600 families have participated. At the end of the program, the foundation supports the municipalities and helps the families to get in contact with local sport organizations and outdoor recreation associations within the local area, so they can continue to be physically active. 

The aim of this study was to evaluate the effects of the controlled pilot intervention on physical activity and sedentary time in children and their families in disadvantaged areas.

## 2. Materials and Methods 

### 2.1. Study Design, Setting and Participants

The study was designed as a controlled pilot study. This pilot study had an explorative approach with a small sample size and was performed in order to trial and evaluate new procedures and logistic feasibility intended for use in a full-scale study. The program had a universal approach, so all children in grade two (8–9 years) and their parents living in a socioeconomically disadvantaged municipality outside Stockholm were invited. The level of high education (>12 years of schooling) was lower in the municipality (women 34% and men 26%) compared to the national average (women 44% and men 35%) [29]. In addition, the municipality had a higher prevalence of overweight and obesity (45% for women and 66% for men) compared to the national average in Sweden (44% for women and 44% for men) [29]. In addition, the municipality was characterized by a high proportion of foreign-born citizens. Children and their parents were recruited through four schools. Families (*n* = 170) from two schools that had already taken part in the program (intervention group) and families (*n* = 87) from two similar schools in the same area that had not yet implemented the program (control group) were invited to participate in the study. Participants in the intervention schools were excluded if anyone in the family had already participated in the program in an earlier school year. In total, 67 children, 58 mothers and 32 fathers agreed to participate in the study. Participant recruitment and retention are illustrated in the flow chart in Figure 1.

### 2.2. The Intervention

The duration of the intervention was 9 months (September 2016 to May 2017). In Figure 2, a logic model is presented for the program with the four intervention components: (1) activity sessions, (2) healthy meals, (3) health information and (4) parental support groups. Motivation and knowledge were identified as possible mediators to children’s and parents’ physical activity and sedentary behavior. The families in the control group were offered the chance to participate in the program after the intervention, one year later. 

#### 2.2.1. Activity Sessions

The families in the intervention group engaged in the activity sessions where different physical activities outside school hours were arranged twice a week, one session on a weekday lasting for two hours and one session on a weekend day lasting for one hour, overall, three hours per week during nine months. The families were invited to try activities such as soccer, basketball, dancing, boxing, outdoor activities and ice-skating. The activities took place at the local school or in the local area and the entire family, including siblings, were engaged. At least one parent had to attend on each occasion. The activities were coordinated by qualified health educators in cooperation with local organizations and outdoor recreation associations. Participation was free of charge and equipment for the specific activity was provided on site.

#### 2.2.2. Healthy Meals and Health Information

The sessions were followed by a healthy, hot meal on weekdays and fresh fruit on the weekends. During the meal sessions the health coordinator discussed different health themes with the families such as the importance of daily physical activity, healthy dietary habits, parental role modeling and parental support.

#### 2.2.3. Parental Support Groups

The parents were invited to four parental support group sessions by qualified external coaches, which were organized by the municipality. The sessions were held during one month, in March, at the same time as the children performed the activity sessions.

All health coordinators use standardized materials developed by the foundation. As a motivation strategy, the families received a stamp in their incentive chart after each activity session and received material to stimulate physical activity after ten sessions. In addition, during the intervention the foundation was in regular contact with the families through text messages and phone calls in order to increase motivation and encourage participation in the activities.

### 2.3. Ethical Statement

All the children and parents gave their informed consent for inclusion before they participated in the study. The study was conducted in accordance with the Declaration of Helsinki and the protocol was approved by the Regional Ethical Review Board in Stockholm, Sweden (Dnr 2016/447-31/2, 2016/1254-32 and 2017/2379-32). Trial registration: ISRCTN11660938.

### 2.4. Outcome Evaluation

All the outcomes were measured in both the children and the parents. The primary outcome was physical activity and the secondary outcome was time spent sedentary. Outcomes were measured at baseline (August/September 2016), and again after 9 months, directly after the intervention (May/June 2017). The measurements were carried out by trained healthcare professionals and researchers in each local school.

#### 2.4.1. Anthropometry

Weight (kg) and height (cm) were measured according to standardized procedures using a calibrated scale to the nearest 0.1 kg and a stadiometer to the nearest mm. BMI was calculated as weight (kg) divided by height (m) squared. Overweight and obesity for children were defined according to the International Obesity Task Force recommendations with different cutoff values depending on age and sex [30]. 

#### 2.4.2. Physical Activity and Time Spent Sedentary

The level of physical activity and time spent sedentary were measured objectively using accelerometry (model GT3X+, Actigraph, LCC, Pensacola, USA) for seven consecutive days. The children and parents were instructed to wear the accelerometers in an elastic belt at the right hip during all waking hours and to remove the monitors for activities involving water.

The software ActiLife Data Analysis, version 6.12.0 was used to process the accelerometer data. The epoch length was set to 5 s for children and 60 s for parents. Non-wear time was defined as 60 min of consecutive zeros allowing for 2 min of non-zero interruptions. Children and parents with at least 500 min/600 min of activity registration per day for a minimum of 1 day were included in the analysis. A time filter was set between 07.00 and 22.59 for the children. The physical activity outcomes were total physical activity (TPA), sedentary time (ST), moderate physical activity (MPA), vigorous physical activity (VPA) and moderate to vigorous physical activity (MVPA). Cut-points for children and parents for sedentary intensity were defined as all activity below 100 cpm/100 cpm, moderate to vigorous intensity was defined as all activity ≥2296 cpm/2020 cpm and vigorous intensity was defined as all activity ≥4012 cpm/5999 pm [31,32,33,34]. The weekly average of ≥60 min per day for children and ≥30 min per day for adults were used in order to categorize the participants into reaching or not reaching the physical activity recommendation.

#### 2.4.3. Socioeconomic Status

The highest level of parental education for the mother or father was self-reported in a questionnaire and used as an indicator of socioeconomic status (SES). The variable was dichotomized into low education (≤12 years of schooling) and high education (>12 years of schooling).

#### 2.4.4. Country of Birth

Parent’s country of birth was self-reported in a questionnaire and categorized as born in “Sweden or other Nordic country”, “Europe” or “outside Europe”.

#### 2.4.5. Process Evaluation

Fidelity to the intervention components was reported by the health educator by documenting the participation of the children and parents in the activity and parental support sessions.

In total, 65 activity sessions were provided for the intervention group (36 children and 47 adults). For those who attended at least once, the mean (SD) participation was 31 (18) times (range: 4–64) for children and 25 (15) times (range: 1–58) for adults. The participation was similar for mothers and fathers, where mothers attended on average 22 times compared to 23 times for fathers. The participation was higher on weekdays compared to weekends for both children and adults. Six children and nine adults participated in the program fewer than 10 times and three children and four adults did not participate at all. In total, 75% of the children and 73% of the adults participated >10 times in the activity sessions and 50% of the children and 30% of the adults participated >50% in the activity sessions. Four parental support sessions were provided and the mean (SD) participation per family in the parental support groups was 2 (1.54) sessions.

### 2.5. Statistical Analyses 

The software Statistica version 13.2 (Statsoft Inc., Tulsa, OK, USA) and IBM SPSS Statistics for Windows, version 25.0 were used for the statistical analyses. Descriptive statistics are presented as means, standard deviations and proportions. An independent samples *t*-test was used for continuous data and a chi square test for categorical data to test for differences between intervention and control groups at baseline. Analysis of variance was performed in order to analyze group differences after the intervention between intervention and control group in TPA, ST, MPA, VPA and MVPA during weekdays and during weekends, adjusted for monitor wear time and baseline values. The continuous outcomes in the analyses were normally distributed and had equal variances between groups for each outcome. Monitor wear time was significant in the analyses and was therefore included in the models. There was an interaction effect between the physical activity variables cpm (*p* = 0.04) and VPA (*p* = 0.05) with sex and group (intervention/control) during the weekends. Therefore, a sub-group analysis was performed with females and males being analyzed separately. A sub-group analysis was also conducted with children and adults separately. The baseline and follow-up values are presented as unadjusted means and standard errors. The differences between intervention and control groups are presented as adjusted means and confidence intervals. All the children and adults that had agreed to participate in the intervention were included in the analysis on an intention to treat basis. Level of significance was set to *p* < 0.05.

## 3. Results

In this pilot study, 67 families participated, 31 children, 29 mothers and 14 fathers in the control group and 36 children, 29 mothers and 18 fathers in the intervention group. In total, six children and 11 adults in the intervention group (20%) and one child and five adults in the control group (8%) dropped out of the study, shown in Figure 1. In total, 66% of the parents had low education and 59% were born outside Sweden. No significant differences in age, sex, parental education, country of birth and anthropometry at baseline were found between the children and adults in the intervention group compared to the control group, as presented in Table 1.

### Physical Activity

In total, 54 of the children (81%) and 73 of the adults (81%) had at least 1 valid day of accelerometer data at baseline and 40 of the children (60%) and 45 of the adults (50%) had at least 1 day of valid data at both baseline and follow-up. The mean (SD) days of wear time was 6.1 (1.2) days for children and 5.7 (1.9) days for adults. In total, 55% of the adults and 59% of the children reached the recommendation of physical activity at baseline.

For the whole group including both children and adults, there was a significant difference in TPA during weekends (*p* = 0.048, mean difference (MD) intervention/control 150 counts per minute) and VPA during weekends (*p* = 0.02, MD 8 min per day) between the intervention and control group, shown in Table 2. There were no significant differences between groups in any of the other physical activity variables or sedentary time during weekdays and weekends.

In the sub-group analyses separated by sex, significant differences were found in TPA during weekends (*p* = 0.03, MD 260 counts per minute), VPA during weekends (*p* = 0.007, MD 14 min per day) and MVPA (*p* = 0.02, MD 21 min per day), during weekends for girls and mothers in the intervention group compared to the control group, shown in Table 3. No significant differences between intervention group and control group were found in any of the physical activity variables and sedentary time for boys and fathers during weekdays and weekends.

In the sub-group analyses separated by children and adults, a statistical difference was found in VPA during weekends (*p* = 0.03, MD 26 min per day) between intervention and control groups for children, shown in Table 4. No differences were found in any of the physical activity variables or sedentary time in adults between the intervention and control groups during weekdays and weekends.

## 4. Discussion

This pilot study evaluated the effects of a family intervention on physical activity and sedentary time in children and their parents from disadvantaged areas. To our knowledge, this is one of the first physical activity interventions that has targeted, engaged and evaluated the whole family in disadvantaged areas. The results showed significant intervention effects in physical activity during the weekends for the whole group, and in particular for girls and mothers. There were no significant intervention effects in physical activity for boys and fathers. There was no intervention effect on the secondary outcome, sedentary time.

The results showed that TPA and VPA increased during the weekends in the intervention group compared to the control group. Furthermore, the girls and mothers in the intervention group increased TPA, VPA and MVPA during the weekends compared to the control group. In the whole group, there was a mean difference between intervention and control groups of 150 cpm in favor of the intervention during the weekends. For mothers and girls, MVPA increased on average 19 min per day during the weekends in the intervention group compared to one minute in the control group. The present results can be compared with a systematic review of physical activity interventions in children where it was shown that MVPA increased on average 4 min per day [23].

Physical activity levels have been shown to be lower for girls compared to boys [7]. Additionally, girls from families with low SES participate less in organized sports during leisure time compared to girls from families with high SES and this is particularly evident for girls with a non-Swedish background [35,36]. It is therefore of great importance to develop successful interventions that are effective for girls from socioeconomically disadvantaged areas. This present intervention succeeded in reaching the girls and the mothers from a socioeconomically disadvantaged area and the physical activity levels increased during the weekends when activity levels have been shown to be lower compared to weekdays. During weekdays a lot of time is structured for work and school so there might not be enough time to engage in physical activity. In previous family interventions, only small effects have been demonstrated on physical activity in favor of the intervention group [27]. In this review and meta-analysis, 19 family interventions were included and it was shown that many previous physical activity interventions within family settings targeted only children with overweight and obesity. Some of the studies also included either the mother or the father in the intervention together with the child. In total, there have been few studies that have engaged and evaluated the whole family and this is particularly true in disadvantaged areas.

In this present pilot study, there was no intervention effect on sedentary time. One explanation could be that the intervention mainly focused on increasing physical activity and not decreasing sedentary time. The results from a previous systematic review suggest that the level of parental involvement was important for the success of the intervention on sedentary behaviors [28].

In some previous universal studies, with similar age groups, positive intervention effects on physical activity have been demonstrated. A 15-week family intervention with extended parental influence during playing time, showed modest changes in self-reported physical activity levels in children (*n* = 24, mean age 7.96 years) but not for parents (*n* = 22) [37]. Similar findings were found in a study by Viitasalo et al. [38], where they showed that a family-based intervention increased self-reported physical activity (+9 min/day in the intervention group and −5 min/day in the control group) and attenuated an increase in sedentary behavior, such as using the computer and playing video games in children. The two-year intervention consisted of 12 counseling sessions and fact sheets for the parents and children aged 6–8 years. Furthermore, the children were encouraged to participate in after school exercise clubs and received financial support for physical activity. Taken together, the positive results from the studies and our present pilot study may be partly due to the level of engagement and involvement of the whole family and that children within this age group may be in a period of life when their physical activity behavior is especially prone to be influenced. However, the results from previous family intervention studies are difficult to compare due to heterogeneity in intervention components, duration of interventions, measurement methods and the grade of involvement of parents.

Although the study may seem small and the effects of the intervention limited, we still find the results encouraging and clinically relevant. The rapidly expanding knowledge on physical activity and health shows that we have underestimated the importance of physical activity for health. With more precise measurements, new prospective studies in children and adolescents show that the benefits in terms of for example effects on important outcomes such as markers of atherosclerosis are greater than we expected [39]. Furthermore, intervention studies in children demonstrate that as little as a 3-min walk has immediate and pronounced positive effects on insulin sensitivity [40]. Decreased insulin sensitivity and hyperinsulinemia are important underlying factors for the increased risk of, not only type 2 diabetes, but also cardiovascular diseases and some common cancers among sedentary individuals.

Furthermore, from a behavioral medicine standpoint, we also find the results encouraging and relevant. Our study includes children and families from socioeconomically vulnerable areas where the barriers to increasing physical activity can be expected to be higher than among children with higher socioeconomic status. Even a slight increase in physical activity after a relatively long follow-up time, can be seen as a positive and important finding.

One major strength of this intervention is the relatively long duration of the intervention and continuous activities every week. In addition, the number of drop-outs was low (15%). Another strength is the objective measurement of physical activity and sedentary time. This present pilot study has also some limitations. The accelerometers cannot register all physical activities and therefore it is not possible to measure for example cycling and water-based activities, which are both popular activities among children. Additionally, the sample size was relatively small and generalizability to other populations may therefore not be appropriate. Given that this is a pilot study, more studies with larger sample sizes are needed. Additionally, children were not randomly allocated to the intervention and control group as the municipality had already decided which schools that should be given the intervention. However, the control schools were matched according to area and educational background of the parents and there were no differences in characteristics between children and parents in the intervention and control groups at baseline. Another potential limitation could be the large variation in the fidelity of participation in the activity sessions for both children and adults and that may have affected the results.

It is of importance that the results from this pilot study demonstrated that the program could positively influence those who need it the most, i.e., girls and mothers from a disadvantaged area, where health needs are the greatest. More studies are needed on how to successfully recruit families from disadvantaged areas and how to improve retention in programs.

## 5. Conclusions

This pilot study shows that it is possible to influence physical activity in families living in disadvantaged areas, in particular in girls and mothers through a family intervention program. This finding may be an important contribution to the further development of effective strategies to promote physical activity in families from socioeconomic disadvantaged areas. Given that this is a pilot study, the results should be interpreted with caution. Further intervention studies are needed with larger study samples.

## Figures and Tables

**Figure 1 ijerph-17-03794-f001:**
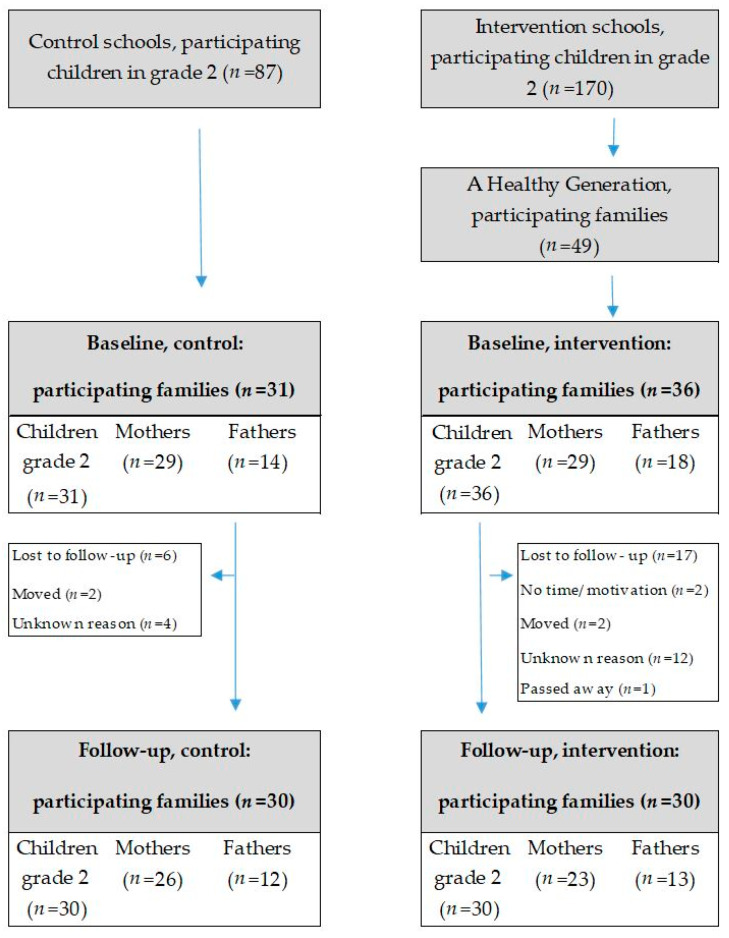
Participant flow diagram.

**Figure 2 ijerph-17-03794-f002:**
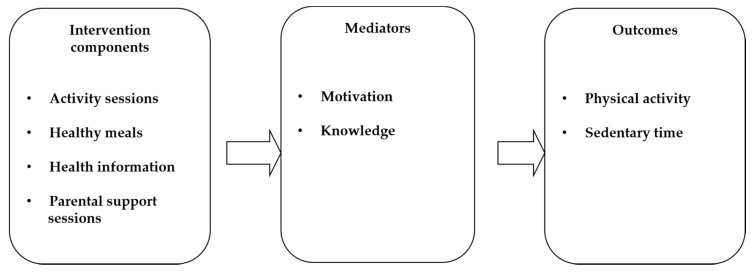
Logic model of the program.

**Table 1 ijerph-17-03794-t001:** Descriptive characteristics at baseline categorized by intervention and control and by children and parents.

Descriptives	Children			Adults		
	Intervention	Control	*p*	Intervention	Control	*p*
	*n* = 36	*n* = 31		*n* = 47	*n* = 43	
	Mean (SD)	Mean (SD)		Mean (SD)	Mean (SD)	
Age (years)	8.2 (0.3)	8.2 (0.3)	0.47	39.5 (6.4)	39.5 (7.2)	0.98
Male (%)	50	61	0.35	38	33	0.57
Low parental education (%) *	68	59	0.46	72	63	0.41
Country of birth, Sweden (%)				43	38	0.67
Anthropometry						
Weight (kg)	27.9 (6.1)	31.2 (8.0)	0.07	81.8 (21.6)	74.9 (15.8)	0.11
Height (cm)	129.8 (6.1)	131.0 (5.9)	0.44	168.9 (8.6)	165.9 (9.8)	0.13
Body mass index	16.5 (2.4)	18.0 (3.7)	0.07	28.5 (7.5)	27.0 (4.7)	0.30

* Low parental education ≤ 12 years.

**Table 2 ijerph-17-03794-t002:** Differences between intervention and control groups in physical activity and sedentary time.

Variables	Intervention			Control			Mean Difference Int-Con * (95% CI)	F	*p*-Value
	*n*	Mean (SD)		*n*	Mean (SD)				
		Baseline	Follow-up		Baseline	Follow-up			
TPA weekdays (counts per minute)	39	490 (194)	528(219)	46	516 (211)	517 (168)	9 (−61 to 79)	0.07	0.80
TPA weekends (counts per minute)	23	409 (173)	603 (421)	40	454 (202)	471 (195)	150 (2 to 298)	4.09	0.048
ST weekdays (minutes)	39	495 (124)	477 (125)	46	468 (104)	479 (91)	−1 (−36 to 35)	0.001	0.97
ST weekends (minutes)	23	490 (143)	423 (75)	40	474 (127)	438 (92)	6 (−29 to 41)	0.11	0.74
MPA weekdays (minutes)	39	35 (15)	38 (19)	46	36 (16)	39 (20)	−2 (−9 to 6)	0.16	0.69
MPA weekends (minutes)	23	25 (17)	31 (16)	40	30 (20)	31 (18)	3 (−5 to 11)	0.48	0.49
VPA weekdays (minutes)	39	14 (11)	16 (11)	46	14 (13)	15 (12)	1 (−3 to 5)	0.12	0.73
VPA weekends (minutes)	23	10 (14)	18 (18)	40	11 (11)	10 (10)	8 (1 to 15)	5.34	0.02
MVPA weekdays (minutes)	39	50 (24)	53 (26)	46	50 (27)	54 (29)	−1 (−11 to 10)	0.02	0.88
MVPA weekends (minutes)	23	35 (23)	49 (29)	40	41 (29)	40 (25)	12 (−0.4 to 24)	3.77	0.06

TPA: total physical activity, ST: sedentary time, MPA: moderate physical activity, VPA: vigorous physical activity, MVPA: moderate to vigorous physical activity. *ANCOVA adjusted for baseline value and accelerometer wear time. SD: standard error, CI: confidence interval.

**Table 3 ijerph-17-03794-t003:** Differences between intervention and control groups in physical activity and sedentary time categorized by sex.

Variables	Intervention			Control			Mean Difference Int-Con * (95% CI)	F	*p*-Value
	*n*	Mean (SD)		*n*	Mean (SD)				
		Baseline	Follow-up		Baseline	Follow-up			
Females									
TPA weekdays (counts per minute)	26	511 (208)	581 (241)	26	471 (181)	481 (158)	65 (−32 to 163)	1.82	0.18
TPA weekends (counts per minute)	15	430 (187)	713 (483)	21	396 (169)	422 (144)	260 (28 to 493)	5.19	0.03
ST weekdays (minutes)	26	483 (126)	450 (136)	26	476 (106)	483 (93)	−20 (−69 to 30)	0.66	0.42
ST weekends (minutes)	15	465 (154)	396 (61)	21	474 (128)	434 (90)	15 (−62 to 33)	0.40	0.53
MPA weekdays (minutes)	26	37 (16)	40 (22)	26	30 (11)	35 (19)	1 (−10 to 12)	0.05	0.83
MPA weekends (minutes)	15	29 (19)	35 (18)	21	24 (16)	25 (14)	7 (−4 to 18)	1.86	0.18
VPA weekdays (minutes)	26	15 (13)	17 (11)	26	11 (10)	12 (11)	4 (−2 to 9)	1.50	0.23
VPA weekends (minutes)	15	9 (10)	21 (21)	21	7 (8)	7 (8)	14 (4 to 24)	8.27	0.007
MVPA weekdays (minutes)	26	52 (26)	58 (29)	26	41 (18)	47 (28)	6 (−10 to 21)	0.48	0.49
MVPA weekends (minutes)	15	37 (26)	56 (33)	21	31 (23)	32 (18)	21 (4 to 38)	6.14	0.02
Males									
TPA weekdays (counts per minute)	13	447 (163)	422 (112)	20	574 (237)	563 (172)	−68 (−150 to 14)	2.89	0.10
TPA weekends (counts per minute)	8	370 (144)	398 (133)	19	515 (220)	525 (232)	−37 (−183 to 108)	0.28	0.60
ST weekdays (minutes)	13	520 (121)	532 (77)	20	457 (102)	475 (90)	37 (−12 to 85)	2.40	0.13
ST weekends (minutes)	8	536 (114)	474 (77)	19	473 (129)	442 (96)	43 (−9 to 96)	2.94	0.10
MPA weekdays (minutes)	13	31 (12)	32 (11)	20	43 (18)	45 (20)	−5 (−17 to 7)	0.64	0.43
MPA weekends (minutes)	8	18 (9)	25 (11)	19	38 (22)	37 (19)	−1 (−16 to 13)	0.03	0.87
VPA weekdays (minutes)	13	13 (8)	12 (10)	20	19 (15)	19 (12)	−2 (−9 to 4)	0.55	0.47
VPA weekends (minutes)	8	13 (20)	12 (12)	19	15 (13)	13 (12)	−1 (−12 to 9)	0.06	0.81
MVPA weekdays (minutes)	13	44 (18)	44 (16)	20	62 (32)	63 (28)	−6 (−19 to 7)	0.91	0.35
MVPA weekends (minutes)	8	31 (18)	36 (13)	19	53 (32)	49 (29)	−0.5 (−19 to 18)	0.003	0.96

TPA: total physical activity, ST: sedentary time, MPA: moderate physical activity, VPA: vigorous physical activity, MVPA: moderate, to vigorous physical activity. * ANCOVA adjusted for baseline value and accelerometer wear time. SD: standard error, CI: confidence interval.

**Table 4 ijerph-17-03794-t004:** Differences between intervention and control groups in physical activity and sedentary time categorized children and adults.

Variables	Intervention			Control			Mean Difference Int-Con * (95% CI)	F	*p*-Value
	*n*	Mean (SD)		*n*	Mean (SD)				
		Baseline	Follow-up		Baseline	Follow-up			
Children									
TPA weekdays (counts per minute)	20	606 (191)	597 (187)	20	693 (185)	635 (102)	−42 (−133 to 49)	0.87	0.36
TPA weekends (counts per minute)	13	429 (204)	727 (522)	17	595 (184)	591 (187)	239 (−63 to 541)	2.65	0.12
ST weekdays (minutes)	20	454 (123)	443 (76)	20	435 (92)	439 (59)	23 (−7 to 53)	2.40	0.13
ST weekends (minutes)	13	505 (182)	421 (70)	17	465 (128)	405 (79)	16 (−32 to 64)	0.48	0.50
MPA weekdays (minutes)	20	43 (13)	41 (10)	20	47 (13)	49 (11)	−4 (−10 to 1)	2.52	0.12
MPA weekends (minutes)	13	29 (16)	235 (15)	17	45 (18)	42 (15)	−1 (−13 to 11)	0.03	0.87
VPA weekdays (minutes)	20	21 (10)	18 (7)	20	25 (12)	20 (8)	−1 (−5 to 3)	0.20	0.66
VPA weekends (minutes)	13	11 (9)	23 (21)	17	19 (10)	15 (11)	26 (18 to 35)	5.55	0.03
MVPA weekdays (minutes)	20	64 (21)	59 (15)	20	72 (23)	69 (17)	−5 (−13 to 3)	1,67	0.21
MVPA weekends (minutes)	13	39 (23)	58 (32)	17	65 (26)	57 (24)	12 (−10 to 34)	1.28	0.27
Adults									
TPA weekdays (counts per minute)	19	367 (100)	454 (231)	26	380 (100)	425 (151)	38 (−70 to 146)	0.51	0.48
TPA weekends (counts per minute)	10	383 (127)	442 (141)	23	348 (142)	383 (151)	30 (−75 to 136)	0.35	0.56
ST weekdays (minutes)	19	539 (112)	513 (155)	26	494 (106)	510 (100)	−23 (−85 to 38)	0.59	0.45
ST weekends (minutes)	10	470 (71)	425 (85)	23	480 (128)	462 (94)	−2 (−59 to 56)	0.005	0.95
MPA weekdays (minutes)	19	27 (13)	34 (26)	26	27 (12)	32 (22)	2 (−12 to 16)	0.06	0.80
MPA weekends (minutes)	10	20 (16)	27 (17)	23	19 (13)	22 (15)	2 (−10 to 14)	0.12	0.73
VPA weekdays (minutes)	19	8 (8)	13 (13)	26	7 (7)	11 (13)	1 (−6 to 9)	0.13	0.72
VPA weekends (minutes)	10	10 (19)	11 (13)	23	5 (7)	5 (8)	6 (−2 to 15)	2.43	0.13
MVPA weekdays (minutes)	19	35 (17)	47 (33)	26	34 (16)	43 (31)	3 (−16 to 22)	0.12	0.73
MVPA weekends (minutes)	10	30 (23)	38 (22)	23	24 (17)	28 (18)	6 (−8 to 20)	0.78	0.39

TPA: total physical activity, ST: sedentary time, MPA: moderate physical activity, VPA: vigorous physical activity, MVPA: moderate, to vigorous physical activity. * ANCOVA adjusted for baseline value and accelerometer wear time. SD: standard error, CI: confidence interval.

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
