# Peer review of "Effectiveness of a Family Intervention to Increase Physical Activity in Disadvantaged Areas—A Healthy Generation, a Controlled Pilot Study"

_ijerph, 2020, doi:10.3390/ijerph17113794_

Round 1

Reviewer 1 Report

This manuscript presents the results of an intervention aimed to increase physical activity in socioeconomically disadvantaged areas in Sweden. The authors posit that a family programme that combines of both active and educational interventions had a significant effect on physical activity in participants. The work appears germane and scientifically sound, and addresses an important issue on public health given that social inequalities associate with poorer physical health in both childhood and adulthood. The paper is well written and well organized. Authors provide essential information about physical health and activity and social inequality within Swedish context. However, the manuscript would benefit from revisions.

Line 115. Authors should clearly clarify if the schedule of the activity sessions was always that way (i.e., one session on a weekday lasting two hours, and one session on a weekend day lasting one hour: overall, three hours per week during nine months).

Line 130. Were parental support group sessions mandatory for the participants? Did at least one parent from each family participate in each support group? At what period of time during the nine-month intervention were the support group sessions held?

Lines 141-147. The information regarding the written consent and ethical approval is duplicated.

Line 160. Please specify the criteria for overweight and obesity.

Line 192. How was fidelity operationalized? For example, were any family excluded from the analyses if they had not participated in at least 80% of the sessions?

Line 290. How many children and families participated in every activity session? How many children and families participated ≥50% of the activity sessions?

Line 348 (Limitations). Authors should discuss the potential limitations of the fidelity variability.

Reviewer 2 Report

The article presents an intervention to change physical activity habits and to decrease sedentarism. The subject of the article is of interest to the journal. Below I make some suggestions to the authors to improve the understanding of the procedures and analysis:

The intervention

Motivation and knowledge were possible mediators of physical activity. Knowledge was acquired through Healthy meals and health information, but how was motivation worked? Was there a motivation strategy for those who attended the sessions less? You could expand on this or narrow it down in the article.
Did language pose a problem with foreign families?

Activity sessions

At least one parent attended the activity sessions. In the case of families with both parents, was it checked that it was not always the same spouse? In what proportion did the two spouses go to the activities? If this was not observed, it could be included as a limitation and be observed in the overall study.
Section 3.2. of the results (Fidelity to the intervention components) should be part of the method. Fidelity to the intervention is an aspect of the method to be monitored by the researchers.

Statistical analysis

Given the sample size, were the data analyzed for parametric testing? (normality, homogeneity of variances).
Indicate the value of the statistical test (t, F, X2,..), not only the p-value.
It is necessary to provide the calculation of the effect size (d Cohen, r, w,..) and to make an assessment.

Discussion
I agree that a major strength of this intervention is the long duration of the intervention, the continuous activities each week, and the low number of drop-outs.
You indicate that participation was higher on weekdays compared to weekends for both children and adults (line 293-294). However, weekly physical activity did not increase. Could you discuss this aspect?
Increasing the physical activity on weekends is one step, but is it enough to improve health? Please discuss.
Do you have any explanation why there was no difference in the group between children and parents? It could be because of the content of the program.

In the discussion, it is necessary to make a more critical assessment with the intervention program. Physical activity rates do not improve during the week. The results of the meta-analysis are necessary but should be contrasted and linked to the results obtained in your study.

The accelerometers were removed in activities related to water and cycling. These activities are intense and can alter the results. Include this limitation within the limitations section (not before). Were any instrument used to collect information on activities carried out without the accelerometer?
